

# Limits and CO2 equilibration of near-coast alkalinity enhancement

Jing He[1,*] and Michael D. Tyka[2,*]

[1]MIT-WHOI Joint Program in Oceanography and Applied Ocean Science and Engineering, Cambridge, MA, 02139, USA
[2]Google Inc., 601 N 34th St, Seattle, WA 98103, USA
[*]These authors contributed equally to this work.

**Correspondence:** Michael Tyka(mike.tyka@gmail.com)

**Abstract.** Ocean Alkalinity Enhancement (OAE) has recently gained attention as a potential method for negative emissions at gigatonne scale, with near-coast OAE operations being economically favorable due to proximity to mineral and energy sources. In this paper we study critical questions which determine the scale and viability of OAE: Which coastal locations are able to sustain a large flux of alkalinity at minimal pH and $\Omega_{Arag}$ (aragonite saturation) changes ? What is the interference distance

between adjacent OAE projects ? How much $CO_2$ is absorbed per unit of alkalinity added ? How quickly does the induced $CO_2$ deficiency equilibrate with the atmosphere ?

Using the LLC270 (0.3deg) ECCO global circulation model we find that the steady-state OAE rate varies over 1-2 orders of magnitude between different coasts and exhibits complex patterns and non-local dependencies which vary from region to region. In general, OAE in areas of strong coastal currents allow the largest fluxes and depending on the direction of

coastal currents, neighboring OAE sites can exhibit dependencies as far as 400 km or more. We found that within relatively conservative constraints set on $\Delta$pH or $\Delta$Omega, most regional stretches of coastline are able to accommodate on the order of tens to hundreds of megatonnes of negative emissions within 300 km of the coast. We conclude that near-coastal OAE has the potential to scale globally to several GtCO$_2$/yr of drawdown with conservative pH constraints, if the effort is spread over the majority of available coastlines.

Depending on the location, we find a diverse set of equilibration kinetics, determined by the interplay of gas exchange and surface residence time. Most locations reach an uptake-efficiency plateau of 0.6-0.8mol $CO_2$ per mol of alkalinity after 3-4 years, after which there is little further $CO_2$ uptake. The most ideal locations, reaching an uptake of around 0.8 include north Madagascar, San Francisco, Brazil, Peru and locations close to the southern ocean such as Tasmania, Kerguelen and Patagonia, where the gas exchange appears to occur faster than the surface residence time. Some locations (e.g. Hawaii) take significantly

longer to equilibrate (up to 8-10 years), though can still eventually achieve high uptake. If the alkalinity released advects into regions of significant downwelling (e.g. around Iceland) up to half of the OAE potential can be lost to bottom waters.

## 1  Introduction

To mitigate the worst effects of climate change, the Paris Agreement aims to limit global temperature warming to below 2°C. This requires not only rapid decarbonization, but also negative $CO_2$ emissions technologies (NET) (Rogelj et al., 2018).

About 150-800 GtCO$_2$ of net negative emissions are needed in the IPCC SSP1-1.9 to SSP1-2.6 scenarios (in addition to



decarbonization) to limit global warming to 2°C by 2100, and this scenario further assumes net negative annual emissions towards the end of the century (Rogelj et al., 2018; Metz and Intergovernmental Panel on Climate Change, 2005; Masson-Delmotte et al., 2021).

On long geological timescales, Earth regulates atmospheric $CO_2$ concentrations by the combined action of surface rock
weathering and ocean $CO_2$ uptake (Penman et al., 2020). High-$CO_2$ conditions lead to elevated temperatures and an intensified hydrological cycle, which increases silicate rock weathering (Archer et al., 2009). The subsequently dissolved alkalinity increases the ocean's capacity for $CO_2$ and the excess atmospheric $CO_2$ dissolves as bicarbonate into the ocean (Zeebe and Wolf-Gladrow, 2001). Indeed the ocean's total dissolved inorganic carbon (DIC) exceeds that of the current atmosphere by 50 fold (Sarmiento and Gruber, 2006).

This mechanism operates on a 10-100ka timescale (Archer et al., 2009), limited by the slow intrinsic kinetics of silicate rock dissolution and the slow introduction of unweathered rock. Exposure of fresh igneous rocks has been linked to rapid cooling of the Earth's past climate (Gernon et al., 2021). Unfortunately, this natural homeostat operates too slowly to mitigate anthropogenic climate change this century. Ocean alkalinity enhancement (OAE) (Renforth and Henderson, 2017) is a proposed approach to accelerate this process in order to increase the ocean's capacity for $CO_2$ and draw down some of the anthropogenic
atmospheric $CO_2$.

The kinetics of rock dissolution can be accelerated in a number of ways. The simplest approach is to increase the rock's surface area through grinding. Powdered rocks such as olivine can then be added to the ocean and will dissolve over the course of years to decades, adding alkalinity (Hangx and Spiers, 2009; Schuiling and de Boer, 2011; Renforth, 2012; Montserrat et al., 2017; Rigopoulos et al., 2018; Meysman and Montserrat, 2017). Alternatively some rocks (e.g. $CaCO_3$) may be prepro-
cessed by calcining, transforming them into more-rapidly dissolving substances such as $CaO$ (Kheshgi, 1995). Major concerns with these approaches are the risk of $CaCO_3$ precipitation (Moras et al., 2021; Hartmann et al., 2022), which would remove alkalinity from the ocean, and the introduction of co-contaminants into the ocean. Iron, abundant in most olivine minerals, could inadvertently fertilize the ocean and cause significant ecological effects (Bach et al., 2019). Silicates would likely shift the phytoplankton species composition towards diatoms (Bach et al., 2019). Less is known about the impact of heavy metals
(Nickel, Chromium, etc) and their effects may be complex (Ferderer et al., 2022).

Finally, changes in turbidity and grinding costs (Li and Hitch, 2015) make deployment of particles <10$\mu m$ impractical, while deployment of coarser particles is limited to shallow waters, as the dissolution is much slower (Montserrat et al., 2017).

An alternative to direct addition of rock mass to the ocean are electrochemical methods which effectively remove acidity from seawater and neutralize it using rocks on land. Acid could be neutralized by using mine tailings and other industrial wastes
or by pumping it into underground basalt formations(Matter et al., 2009; McGrail et al., 2006; Goldberg et al., 2008). Several variants has been proposed based on electrolysis (House et al., 2007; Rau, 2009; Davies et al., 2018) or bipolar electrodialysis (Eisaman et al., 2018; de Lannoy et al., 2018; Digdaya et al., 2020), all essentially producing either pure NaOH or a basified seawater stream, which would be returned to the ocean to increase the pH and elicit $CO_2$ drawdown. The disadvantage is the significant electrical energy requirement and the fact that the produced alkalinity is relatively dilute ( 1mol/kg (de Lannoy
et al., 2018)), exacerbating transport costs out to sea, compared to shipping powdered rock. Prior assessments of shipping costs



(Renforth, 2012) when using dedicated fleets have focused on transport of rock-based, solid alkalinity (notably olivine), which have a high molality of alkalinity ( 25mol/kg).

Regardless of the alkalinity source, OAE methods leverage the marine carbonate system (Renforth and Henderson, 2017), a multiple equilibrium state (Zeebe and Wolf-Gladrow, 2001) described by the equation

$$CO_{2(atm)} \rightleftharpoons CO_{2(aq)} + H_2O \rightleftharpoons H^+ + HCO_3^- \rightleftharpoons 2H^+ + CO_3^{2-} \tag{R1}$$

Dissolved inorganic carbon (DIC) is the combined concentration of all carbonate moieties. Addition of alkalinity (e.g. $OH^-$) shifts the above equilibrium to the right by consuming $H^+$ ions, thus lowering the partial pressure of $CO_2$ in the ocean and driving further ocean $CO_2$ uptake (Middelburg et al., 2020; Zeebe and Wolf-Gladrow, 2001). As the sea-surface $CO_2$ exchange is rate limiting (surface water experiences an equilibration time scale on the order of weeks to years (Jones et al., 2014)), the
addition of alkalinity causes a local increase in pH and aragonite saturation ($\Omega_{Arag}$) which affect the local ecology (Bach et al., 2019). Furthermore, increases in aragonite saturation could lead to precipitation of calcium carbonate, which removes alkalinity from the surface water and is counter productive with respect to $CO_2$ uptake (Moras et al., 2021).

A number of previous studies have used ocean circulation models combined with a carbon cycle model to estimate the carbon uptake potential of various hypothetical OAE scenarios (Köhler et al., 2013; González and Ilyina, 2016; Feng et al.,
2017; Ilyina et al., 2013; Keller et al., 2014; Burt et al., 2021; Tyka et al., 2022) Some of these studies investigate very high rates of alkalinity injection to test the limits of OAE. Ilyina et al. (2013) simulated alkalinity addition on the order of 2.8Pmol/yr (for an approximate uptake of 50 GtCO$_2$/yr). González and Ilyina (2016) added enough alkalinity to remove around 44GtCO$_2$/yr. Both these studies found drastic changes in pH and the carbonate saturation state.

Most of these simulations consider globally uniform alkalinity injection patterns, which is unrealistic for practical deploy-
ment and provides little insight into which geographical locations are ideal for conducting OAE. An ideal region (for purposes of negative emissions) minimizes the effect of added alkalinity on the local carbonate system and ecology, while maximizing the $CO_2$ uptake per unit alkalinity added.

Several authors have conducted scenario-driven and locally-resolved simulations. Köhler et al, 2015 investigated finely ground olivine addition from ship tracks for a total uptake of 3.2 GtCO$_2$/yr, imagining the distribution of alkalinity via ballast
water of commercial ships. These ship tracks span the full ocean extent from 40°S to 60°N, although heavily weighted to the area between 20°N to 50°N.

Feng et al. (2017) simulated adding olivine along global coastlines where continental shelves are shallower than 200 m. They found that to stay below aragonite saturation levels of $\Omega_{Arag}$=3.4 and $\Omega_{Arag}$=9, coastal olivine addition can remove around 12 GtCO$_2$/yr and 36 GtCO$_2$/yr, respectively. Some finer-grained studies have been undertaken: Burt et al. (2021) tested regional
alkalinity addition based on eight hydrodynamic regimes in a 1.5° model, and Tyka et al. (2022) simulated alkalinity addition at individual points in a 3° lat-lng grid. Both studies revealed that the pH sensitivity and the efficiency of $CO_2$ uptake vary geographically and temporally.



Here, we also study alkalinity addition through a practical and economic lens, focusing on electrochemical methods, which produce NaOH or other rapidly-dissolving forms of alkalinity. We begin with the assumption that the optimal places for electrochemical alkalinity production would be on the coast, with access both to seawater and low-cost renewable electricity. To minimize risks to coastal ecosystems and ensure adequate spreading and quick dilution, the alkalinity would be transported some distance off-shore. This is increasingly critical for larger scale deployments, to avoid high concentrations of alkalinity. We wish to determine how far off coast and over what area alkalinity can be added to the surface ocean while staying within conservative biological and geochemical limits. While these issues are less relevant to initial small-scale OAE our goal is to examine the limits of the technology's potential scale. A judicious amount of OAE may also be beneficial, by stabilizing or reversing the anthropogenic acidification of the surface ocean (Albright et al., 2016; Feng et al., 2016). Specific implementations of OAE may also be subject to additional limitations such as trace metal contamination (Bach et al., 2019), which we do not address here.

Increases in alkalinity (and subsequent increases in DIC) change the activities of all chemical moieties involved in the carbonate system (Middelburg et al., 2020; Zeebe and Wolf-Gladrow, 2001), many of which are relevant to marine organisms (Riebesell and Tortell, 2011). Both the direct impact on marine species and the risk of triggering calcium carbonate precipitation must be considered (Bach et al., 2019; Hartmann et al., 2022). Given the complexity of the carbonate system and the variety of responses to each parameter there is no single "correct" choice of proxy (Fassbender et al., 2021) by which to quantify the shift in carbonate state, although the parameters are strongly correlated with each other. Further, what constitutes a safe limit for any given ocean parameter is under debate and likely varies significantly between regions, thus a blanket hard limit is difficult to establish. Here we use two proxies to quantify changes in the carbonate system: $\Delta$pH and $\Delta\Omega_{Arag}$.

Prior studies simulated the addition of uniform amounts of alkalinity over some defined area and measured the varying response of ocean parameters. However, because the sensitivity of these parameters varies over more than an order of magnitude we designed our experiment in reverse, i.e we adjust the alkalinity addition rate in each grid cell to result in a uniform and relatively small change of a given parameter. We can then examine how the injection rate varies and construct maps that indicate regions of high suitability for OAE.

Finally, the effectiveness and timescale of $CO_2$ uptake due to an OAE deployment in a given region is of interest we can define the uptake efficiency $\eta_{CO2}$ as

$$\eta_{CO2}(t) = \frac{\Delta DIC(t)}{\Delta Alk}, \tag{1}$$

where $t$ denotes the time since alkalinity was added. Following the addition of some quantity $\Delta Alk$ to seawater, the ocean will begin taking up $CO_2$, eventually reaching a maximum $\eta_{CO2}(t = \infty) \approx 0.8$ (depending on the exact state of the carbonate system) (Renforth and Henderson, 2017; Tyka et al., 2022). However, the equilibration kinetics of this equilibration are known to vary spatially due to differences in the gas exchange timescales and the surface residence time of $CO_2$ deficient water (Jones et al., 2014; Burt et al., 2021). We thus conducted simulated experiments with short, localized pulse injections, followed by tracking of the total excess alkalinity and DIC relative to a reference simulation as done previously with a much coarser model



(Tyka et al., 2022). This gives an accurate picture of where alkalinity from a particular injection point is advected to, how much alkalinity is lost to the deep ocean and how much and when $CO_2$ uptake can be expected.

## 2 Methods

### 2.1 The Model

We use the ECCO (Estimating the Circulation and Climate of the Ocean) LLC270 physical fields (Zhang et al., 2018), to simulate the transport of alkalinity by currents and model alkalinity addition in near-coast areas globally. We inject alkalinity to the simulation in strips along all global coastlines, 37 km wide and larger. ECCO is an ocean state estimate based on the MIT general circulation model (MITgcm) (Marshall et al., 1997) that also integrates all available ocean data since the onset of satellite altimetry in 1992. ECCO uses the adjoint method to iteratively adjust the initial conditions, boundary conditions,

forcing fields, and mixing parameters to minimize the model-data errors (Wunsch et al., 2009; Wunsch and Heimbach, 2013). This produces a three-dimensional continuous ocean state estimate that agrees well with observational data. We use the LLC270 configuration with a 1/3° degree horizontal resolution (Zhang et al., 2018). All input and forcing files needed to reproduce the ECCO state estimates and the source code are freely available online, and we use them to reproduce the LLC270 flow fields. The LLC270 configuration uses a lat-lng-cap (LLC) horizontal grid, which uses 5 faces to cover the globe. The horizontal

resolution ranges from 7.3 km at high latitudes to 36.6 km at low latitudes, and has 50 vertical layers whose grid thickness goes from 10m near the ocean surface to 458m at the bottom (Zhang et al., 2018). We use the iteration-42 state estimate described in Carroll et al. (2020), which spans the years 1992-2017.

To represent the ocean carbonate system, we used the gchem and dic packages within MITgcm. The ocean carbon model was based on Dutkiewicz et al. 2005 and uses 5 biogeochemical tracers (DIC, Alkalinity, phosphate, dissolved organic phos-

phorus, and oxygen) to simulate the movement of total dissolved inorganic carbon (DIC) within the ocean. In this model, DIC is advected and mixed by the physical flow fields from the MITgcm, and the sources and sinks of DIC are: $CO_2$ flux between the ocean and atmosphere, freshwater flux, biological production, and the formation of calcium carbonate shells. The biogeochemical tracers were initialized with contemporary data from GLODAPv2 mapped climatologies (Lauvset et al., 2016; Olsen et al., 2017) where possible, or using data from Dutkiewicz et al. (2005) and were allowed to relax locally by running 100yrs

of forward simulation (looping the ECCO forcing fields). Atmospheric $CO_2$ concentrations were held constant at 415µatm, rather than trying to anticipate future emission scenarios. The surface carbonate tracers were found to stabilize during this time.

Since we are not simulating a full Earth system, our model does not account for feedbacks of other carbon sinks which reduce the impact of moving $CO_2$ from the atmosphere to the ocean (Keller et al., 2018). Wind speeds, used to calculate the gas exchange, are imported from the LLC270 forcing data. To simulate ocean OAE, we forced the simulation by adding pure

alkalinity to the surface ocean in specified locations to the top grid cell (10m depth) and at a parameterized rate; this assumes that alkalinity is of an effectively instantly dissolving nature, such as an NaOH solution. This avoids complicating factors arising from slower-dissolving materials such as fine olivine powder, for which dissolution rates vary with ocean conditions and may sink out of the surface layers before complete dissolution (Fakhraee et al., 2022). We focused on alkalinity addition in



coastal bands following shorelines because that is economically most viable and accessible for shipping or pipelines. Feedbacks
of elevated alkalinity on the rate of surface calcification are also not explicitly modeled.

Six coastal strips are examined with widths of approximately 37, 74, 111, 185, 296 and 592 km, although it varies slightly
due to the varying grid cell sizes in the LLC270 grid. Feng et al. 2017 used a coarser 3.6° lon by 1.8° lat model, and their
injection pattern roughly corresponds to our 296 km coastal strip. The much finer LLC model allows us to resolve coastal
features in greater detail and to test thinner injection strips. We also examined injection in discrete locations spaced 200 km or
400 km apart along the coastline, in circular patches  120km wide.

All runs presented in this paper use the same approach: First a reference simulation is run (spanning 20 ECCO years 1994-
2019). Then a second run is conducted starting at the same starting conditions, with an alkalinity forcing added, which perturbs
the system in some way. We then analyze the difference in the carbonate systems ($\Delta$pH, $\Delta$Omega, $\Delta$DIC, etc.) between
these two runs. Since the carbonate model does not influence the flow field, there is no divergence in the flowfields over the
25 simulation years and the two trajectories are directly comparable. These simulations are run on a small MPI cluster (13
machines, 59 processes each) on Google Cloud Engine, and take about 6 hours of wall time per simulation year.

## 2.2  pH and Omega limits

The carbonate chemistry in different regions varies in its sensitivity to alkalinity injection, owing to local differences in ocean
circulation, gas exchange and carbonate chemistry. The goal of our experiments is to determine the maximal alkalinity addition
rate which can be sustained at any given grid point which limits the change in one of two surface parameters, pH and the
aragonite saturation $\Omega_{Arag}$ to some chosen value.

Here we chose target constraints $\Delta$pH$_{tgt}$=0.1 or $\Delta\Omega_{tgt}$=0.5. These values are somewhat arbitrary and serve simply to calcu-
late the relative sensitivity of different regions. However, as an intuitive point of reference, the already incurred anthropogenic
surface acidification since preindustrial times (Doney et al., 2009) is $\Delta$pH≈-0.1. Likewise a change of $\Delta\Omega_{Arag}$=+0.5 is un-
likely to trigger carbonate precipitation according to Moras et al. (2021) who established an $\Omega_{Arag}$ threshold of 5.

In practice which limits are acceptable is subject to debate and likely different in different locations and we do not attempt to
anticipate them here, focusing merely on the relative capacity of different ocean regions with respect to these ocean parameters.
Our approach is as follows. Each surface gridpoint that is part of the coastal injection strip is given a particular baseline injection
rate $r$ (in mol/m$^2$/s). At every timestep and for every gridpoint, the local (in time and space) $\Delta$pH is calculated using the
carbonate model and a reference value obtained from an unperturbed reference simulation ($\Delta$pH = pH - pH$_{ref}$). If this value
is lower than $\Delta$pH$_{tgt}$ then alkalinity is added according to the baseline rate, if not then addition is skipped for this timestep.

This mechanism is insufficient to ensure the pH does not exceed the maximal value, as the change in pH is determined
not only by the local alkalinity addition, but also by advection of alkalinity from neighboring cells and seasonally varying
biological activity. We thus iteratively adjusted the baseline rate for each grid cell to empirically determine a rate which gives
rise to approximately the desired $\Delta$pH in the following way.

First a pilot simulation was run where the baseline rate was set uniformly to an extreme value of $r$=400 mol/m$^2$/yr, (higher
than any region can accommodate). We ran this simulation for 3 years and recorded the observed amount of addition at each





grid cell (generally much lower than the baseline as the above algorithm prevents excessive addition). Second, we reran the simulation using a new, position-dependent baseline rate calculated from the amounts actually added from the pilot simulation using a linear extrapolation to our desired pH maximum. We ran this second simulation for 8 years. We found that the observed $\Delta$pH is now generally very close to the desired $\Delta$pH$_{tgt}$, however some regions still exceed the target value while others undershot. We thus performed a third simulation where we adjusted the addition rate at any grid point inversely proportional to the observed pH deviation yielding a final third simulation which was allowed to run for 20 years using the ECCO forcing fields from 1995 to 2014. We found that this procedure yielded a relatively narrow distribution of $\Delta$pH or $\Delta\Omega_{Arag}$ for all grid points in the injection strip although some variability remains (Fig. S1). A separate iterative optimization was performed for every injection pattern. Grid points outside of the injection strip showed much smaller changes in pH and never exceeded the target $\Delta$pH. The same procedure was used in a separate set of experiments for $\Delta\Omega_{Arag}$.

Once the rate of OAE is stable and acceptable, we can measure how much alkalinity is being added at each grid point. Note that because of the considerable interdependence between nearby grid points there is no one unique injection pattern that satisfies the $\Delta$pH or $\Delta\Omega_{Arag}$ condition, however multiple independent optimization runs started at different ECCO years yield injection patterns that match very closely.

## 2.3 Pulse additions

When alkalinity is added to the surface ocean it lowers the partial pressure of $CO_2$ (pCO$_2$) and thus increases the rate at which $CO_2$ dissolves in the surface ocean. The effectiveness $\eta_{CO_2}$ of this uptake is determined by a number of factors which vary significantly by location. The timescale of gas exchange $\tau_{CO_2}$ is approximately 3-9 months and varies by location (Jones et al., 2014) while the residence time $\tau_{res}$ of water parcels in the mixed layer varies over shorter time, between 2 and 20 weeks.

Thus the resultant equilibration efficiency ratio $\tau_{res}/\tau_{CO_2}$ was found to be significantly below 1.0 in 95% of ocean locations (Jones et al., 2014). However, $CO_2$-deficient water parcels initially lost from the mixed layer can re-mix into the mixed layer at some later time and thus drive further equilibration elsewhere and over longer timescales. This longer term effect was not explicitly modeled in previous work (Jones et al., 2014) and results in a complex equilibration curve which is not well captured by a single exponential function. As the kinetics of this longer-term equilibration depend on the deep transport and mixing of the lost alkalinity it has to be simulated explicitly.

We thus extend the work of Jones et al. (2014) by simulating pulse injections of alkalinity in a variety of locations using the ECCO flow fields. These simulations thus explicitly include all the relevant aspects together (gas-exchange, Revelle factor, surface transport, mixed layer-depth, residence time and remixing), by measuring the actual excess $CO_2$ uptake of the ocean relative to the unperturbed reference simulation. However, because the alkalinity is also distributed horizontally over great distances and mixes from different origins it is impossible to disambiguate the $CO_2$ uptake timescale of different injection points from a single simulation. One solution to this problem is to use a Lagrangian approach (van Sebille et al., 2018) which allows for the tracking of stochastic particles. Here we chose a simpler approach. For a select number of coastal locations we run a separate simulation and inject a 1-month pulse of alkalinity. Following the pulse we monitor the total excess DIC in the ocean relative to a reference simulation, the distribution of alkalinity across the depth layers and the pCO$_2$ deficit at the surface





over time. Ideally the length of the pulse would be a single timestep, however this would either necessitate an extreme addition rate or a tiny total quantity of added alkalinity, which would lead to a poor signal to noise ratio during analysis. The choice of pulse length thus represents a compromise, as this length is still much shorter than the overall relaxation time. Ideally such a

pulse injection experiment could be conducted for every grid point (as was done with a coarse model in Tyka et al. (2022) and at different times of the year, however, because each pulse requires a whole separate simulation, this exceeded our computation capacity with the high-resolution ECCO model. Thus we chose 17 individual locations of interest along most major coastlines with pulses occurring in January.

## 2.4   Alkalinity injection from ships

In addition to the steady-state perturbation of ocean parameters over large areas of OAE deployment, it is critical to examine the short term impacts that arise right at the injection site which will temporarily take the local carbonate system into an extremely alkaline regime. This is unlikely to be a concern for gradually dissolving alkalinity such as ground olivine (Hangx and Spiers, 2009; Schuiling and de Boer, 2011), but highly relevant for rapidly dissolving alkalinity such as NaOH solution or other solubilized alkaline media.

Of interest is the dilution speed of the added alkalinity (we assume here a solution of 1M NaOH) against the timescale at which homogeneous nucleation of aragonite is triggered and needs to be avoided.

The most natural approach, used also in waste disposal, would be to inject directly into the turbulent wake of the ship to mix the discharge with seawater as rapidly as possible (Renforth and Henderson, 2017). The dilution kinetics have been studied and modeled in previous work (Chou, 1996; IMCO, 1975) and large discrepancies exist between the published models. The

IMCO model uses the following empirical form for the unitless dilution factor as function of time $t$

$$D(t) = \frac{c}{Q} U^{1.4} L^{1.6} t^{0.4} \qquad (2)$$

where t is time (seconds), Q is the release rate (m$^3$/s), U is the speed of the ship (m/s), L is the waterline length (m). C is an empirical constant, set to 0.003 for release from a single orifice and 0.0045 for release from multiple ones. Intuitively larger speeds and longer ships have more turbulent wakes producing faster dilution. Chou (1996) used the following, similar model

which instead of waterline length uses the width of the ship, B, in units of meters, to account for the ship size.

$$D(t) = \frac{0.2108}{Q} U^{1.552} B^{1.448} t^{0.552} \qquad (3)$$

Chou's formula gives much faster dilution rates and was verified against field testing data. Other work (Lewis, 1985; Byrne et al., 1988; Lewis and Riddle, 1989) used even higher exponents on the time t so we take the IMCO formula to be an upper limit though in general no universally applicable law can be expected as the dispersion timescale will inevitably depend on local

conditions. For a given starting concentration $C_0$ of the alkaline effluent (e.g. 1mol/L NaOH) we can calculate the resultant alkalinity by considering the dilution with seawater with alkalinity $Alk_0$



$$Alk(t) = \frac{1}{D(t)}C_0 + \left(1 - \frac{1}{D(t)}\right)Alk_0 \tag{4}$$

We can then determine the pH(t) and the carbonate saturation state $\Omega(t)$ by solving the carbonate system at any given $Alk(t)$. We used PyCO2SYS (Humphreys et al., 2020) to solve the carbonate system numerically, assuming PyCO2SYS default values

and $Alk_0$=2300mol/m$^3$ and DIC=2050mol/m$^3$. These analytical models are valid only on timescales smaller than one hour after which dilution kinetics are driven by the local background turbulence rather than the immediate influence of the wake turbulence. Thus longer-scale dilution effects will vary substantially from location to location. As the timescale considered here is much shorter than the typical $CO_2$ gas-exchange timescale we do not explicitly model $CO_2$ uptake during this initial dilution.

## 2.5 Estimation of transport costs

Alkalinity prepared on land must be transported out to sea, which incurs costs. While at very small scale it could be released right at the coast, for larger overall OAE deployment the alkalinity will need to be spread over greater areas (to avoid excessive local pH or Omega changes) which increases the cost for every additional unit of alkalinity added. Having obtained maps of the allowable rate of OAE in any given area we can estimate the transport costs for each of our simulated scenarios. Large-scale

maritime shipping costs are around \$0.0016-\$0.004 t$^{-1}$km$^{-1}$ (Renforth, 2012). For each grid point where alkalinity is injected, we calculate the distance $D$ to the nearest coast, and take double that to be the minimum round trip distance for a ship to travel. Realistically, a ship will have to travel farther than D since it needs to go to the nearest port or NaOH factory, so this is the lower bound on the transport distance. This allows us to calculate the lower bound of the shipping cost per tCO$_2$ for each grid point. The scenario model provides the alkalinity injection rate for each grid point, and assuming an eventual uptake efficiency

$\Delta_{CO_2} \approx 0.8$ we can obtain the total shipping cost for every grid point. Summing the total cost over all grid points in which injection occurs and dividing by the total expected global $CO_2$ uptake yields a lower bound of the average effective global transport cost per tonne $CO_2$.

## 3 Results

### 3.1 Injection capacity

In all our simulations the alkalinity flux in the injection grid cells was iteratively adjusted to elicit a change in pH or the change in Omega (though not simultaneously) to a value of either $\Delta$pH=+0.1 or $\Delta\Omega_{Arag}$=+0.5 respectively. Due to the correlation between neighboring grid cells, seasonal and year-to-year changes in currents and biotic activity these constraints are not perfectly satisfied, but we were able to confine them to a narrow range around the desired value (Fig 1e, Fig S1a). The injection rate, as well as $\Delta$pH, $\Delta\Omega_{Arag}$, stabilize within the first 5-6 years of the simulation and remain stable for the remainder of

the simulation (Fig. 1g and Fig. S1), indicating that a steady state is reached where the alkalinity addition rate is matched by outflowing alkalinity into open ocean areas and by neutralization by atmospheric $CO_2$. Note that while the time-averaged $\Delta$pH





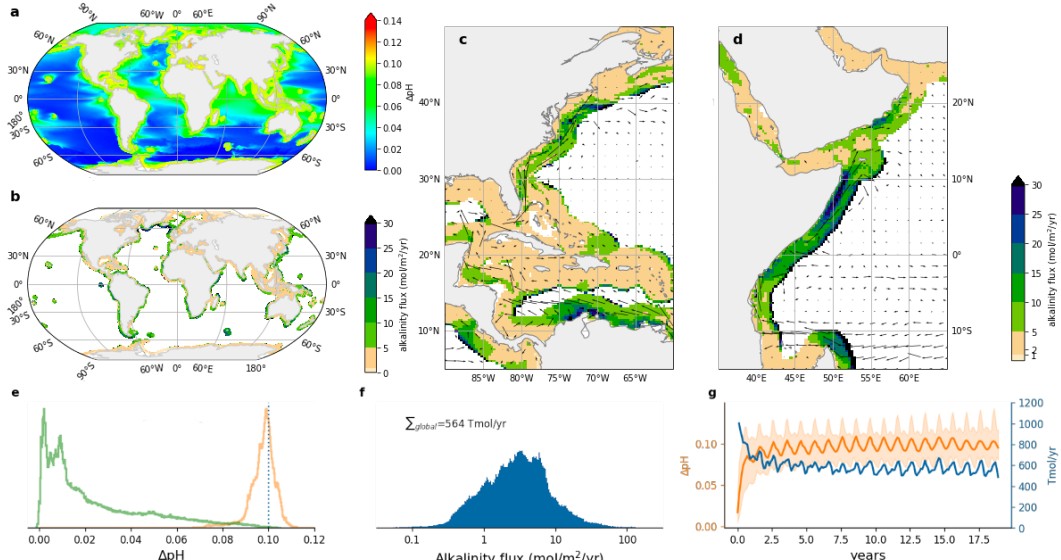

**Figure 1.** Shown is a coastal injection in a strip 296 km wide, subject to $\Delta pH_{tgt}$=0.1. All time-averages ran over years 5-20. (a) Averaged pH change compared to a reference simulation with no added alkalinity. (b) Average alkalinity flux at each grid point. from the coast. (c) and (d) Injection flux for two example regions. Annually averaged surface currents are overlaid as a vector field. The variation of sustainable alkalinity flux in different parts of the coastal strip is apparent. (e) Normalized histograms of the annually averaged $\Delta pH$ over all grid points in the injection strip (orange) or outside the strip (green). The dotted vertical line indicates $\Delta pH_{tgt}$=0.1. (f) Distribution of the sustained alkalinity flux in the injection grid points. Note the x-axis is log-scaled, showing that injection flux spans >2 orders of magnitude. The total global total injection rate is shown above the histogram. (g) Total global alkalinity addition rate (summed over all grid points in the coastal strip) in blue and median pH change from the reference simulation (exclusively over grid points in the strip) in orange. The shading shows the 10th and 90th percentile range. The addition rate and pH change stabilize after 5 years.

is close to 0.1, there is significant temporal variability that leads to $\Delta pH$ slightly exceeding 0.1 at some parts of the year (Fig. S1).

As expected, $\Delta pH$ or $\Delta\Omega_{Arag}$ outside the injection grid cells is much lower and never exceeds the target value. However, the effect on adjacent areas outside of the injection grid points is variable and depends on the pattern of ocean currents that sweep alkalinity away from injection areas. For instance, western boundary currents carry the coastal excess alkalinity far out into the open ocean, so we see elevated changes in the North Atlantic and Indian oceans, even outside the injection areas.

For each injection pattern and chosen limit we can now obtain a global map of steady-state alkalinity flux (mol/m$^2$/yr), which shows the variability of capacity for OAE (Fig 1 b,c and d). We note substantial variability on multiple scales. Firstly on a large scale, some coastal areas have a fundamentally greater capacity for distributing and neutralizing alkalinity flux than others. Large capacities are found around islands which sit in or near ocean currents, as those rapidly sweep the alkalinity away from near-coast areas. Examples include Kerguelen, Easter Island and Hawaii. Continental coasts which exhibit large capacities are found around south and east Africa, off the coast of Peru and Brazil, south-east Australia and the west coast





of Japan. Finally an area of very large tolerance for alkalinity addition is found in the northern Atlantic, however, as will be
shown later, this is due to downwelling and deep water formation, which is highly undesirable for OAE as the alkalinity cannot
efficiently equilibrate with the atmosphere before being lost. Conversely inland seas and partially enclosed seas exhibit the
smallest capacity for OAE, notably the Red Sea, the Mediterranean and the Baltic Sea. An interesting counterexample is the
Gulf of Mexico and the Caribbean Sea which, owing to the traversing Gulf Stream, have significant capacity for OAE.

As expected, the large scale patterns obtained by limiting $\Delta pH$=0.1 or $\Delta\Omega_{Arag}$=0.5 are very similar (Fig. S2) up to a linear
factor. We note however that actual absolute limits for these two metrics may differ significantly. From the perspective of
preventing precipitation due to excessive $\Omega_{Arag}$ the available headroom is much smaller at tropical latitudes, where $\Omega_{Arag}$ is
already close to 4, than near the poles where $\Omega_{Arag}$ is as low as 1-2(Lauvset et al., 2016; Olsen et al., 2017). Likewise for pH
the actual ecologically tolerable limits may vary from coast to coast, and we do not attempt to anticipate them here. We note
however that for our examined constraints (both of which are very conservative) a significant amount of negative emissions can
be obtained even in very narrow coastal strips as the transport out to open sea is very efficient. This obervation is consistent
with prior work by Feng et al. (2017).

On a fine scale we find that the sustainable injection flux varies over 2-3 orders of magnitude between nearby grid points
(Fig. 1f) with a distribution that is approximately log-normal. The variance is even larger for thin injection strips (Fig. S1)
and very large fluxes can be sustained in some locations if the ventilation out of the strip is high enough. Depending on the
prevailing currents, the highest injection rate can both be found on the outside of the injection strip or on the inside.



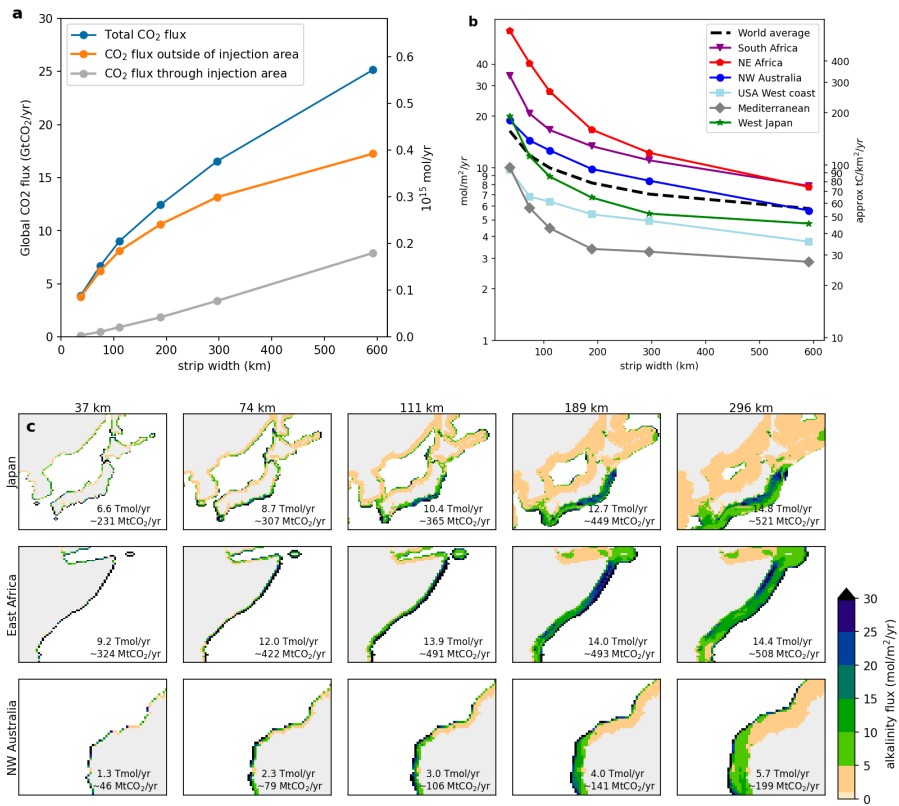

**Figure 2.** (a) Total $CO_2$ flux (GtCO$_2$/yr), subject to $\Delta pH_{tgt}$=0.1, broken down into flux through the injection area and outside of the injection area. The total OAE potential is notably sublinear with increasing strip width. For narrow strips the total $CO_2$ flux occurs almost entirely outside of the injection area. (b) Dependence of the averaged injection flux (mol/m$^2$/yr) on the width of the coastal injection strip for different coastal regions and strip widths. A large range of injection capacities is observed. In general regions with strong coastal currents (e.g. East Africa) are able to receive an order of magnitude more alkalinity per unit area than enclosed seas (e.g Mediterranean). The black dashed line averages all coastal regions. Widening the strip area reduces the average injection flux and explains the sublinear total OAE potential observed in (a) (c) Injection flux (mol/m$^2$/yr) shown for different strip widths in 3 different regions (from top to bottom: Japan, East Africa and Northwest Australia). The inset text indicates the total alkalinity added per year in the shown area. Widening strips allow more alkalinity to be added overall, however saturation of near-coast areas occurs in most regions (esp evident in Northwest Australia). The largest injection flux is often found directly at the strip boundary, owing to easy diffusion out to open sea. However, it is not always the case that the highest injection flux occurs at the strip edge, as seen in the Japan and East Africa examples. These findings illustrate the highly non-local nature of the injection capacity.



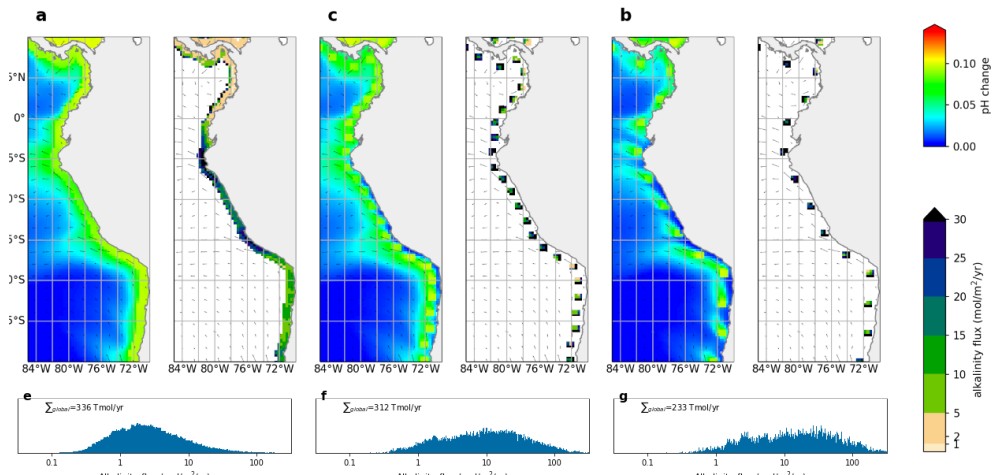

**Figure 3.** Alkalinity enhancement in different patterns, exemplified at the west coast of South America: (a) Injection in a contiguous strip (b) Injection in 200 km separated patches (c) Injection in 400 km separated patches (e)-(g) Distribution of alkalinity flux (globally) for the same scenarios as exemplified in (a)-(c). The total global OAE rate is shown inset. Separation of the injection strip into 200 km patches leads to higher spot injection rates but barely reduces the total injection rate. The pH influence on coastal waters bleeds into neighboring patches. Separation by 400 km does not significantly raise spot injection rates further, but total injection rate is reduced. The pH influence is now much more separated.

For all regions we observe that while widening the injection strip increases the total allowable rate of alkalinity injection, the increase is sublinear. Every additional unit of alkalinity added needs to be transported further offshore and the average injection rate decreases per unit area. This is consistent with the view that the majority of the neutralization of the added alkalinity by invading $CO_2$ occurs outside the injection strip and the local pH is primarily determined by the rate of transport of alkalinity

into other areas (Burt et al., 2021). This is confirmed by integrating the total $CO_2$ flux over the injection strip and over the rest of the ocean surface, relative to the reference simulation. Figure 2a shows that especially for thin coastal injection strips, direct gas exchange through the strip surface accounts for only a minor component of the induced $CO_2$ uptake and the majority occurs outside of the injection areas. As the strip widens however, this fraction increases significantly. Especially in regions with weak transport we observe that often a larger quantity of alkalinity can be added right at the border of the strip than in

the middle, as alkalinity can dilute to bordering areas that are not directly receiving alkalinity (Fig. 2c). Indeed, the largest alkalinity fluxes observed occur when the strip widths are very thin (Fig. S1). The consequence is that any particular coastal region can increase its capacity for injection by going further out to sea, but that there are diminishing returns of doing so, i.e. the increase in capacity is sublinear with width. The influence of widening the injection area is also shown in detail for three regions (Japan, Northwest Australia, and eastern Africa) in Figure 3c and a larger number of regional details are included in

the supplementary material in Figure S4.



In general we find that the sensitivity of pH and $\Omega_{Arag}$ at any given grid point is highly dependent on the surrounding pattern of injection. We conducted two additional experiments in which rather than a contiguous strip, injection occurred at discrete points placed either 200 km or 400 km apart. At 200 km apart we observe much higher injection fluxes in each injection patch, but the total global injection flux barely changes (Fig. 3). Placing injection patches 400 km apart instead of 200 km apart did not

further increase the sustainable flux in each injection patch, and reduced the overall injection capacity by 25%. This suggests that at 200 km there is still significant cross-correlation between neighboring patches, which is apparent when looking at the pH changes observed: the pH impact bleeds into neighboring patches (Fig. 3b). At 400 km apart, however, there is much less interference between adjacent patches and the injection limits are simply dominated by local current patterns (Fig. 3c). These observations are consistent with prior work (Jones et al., 2012) which found a global median pCO$_2$ autocorrelation length

is about 400 km. Thus in order to maximize any particular coastline's injection potential, injection areas should be placed at most 200-400 km apart. The location of ports, infrastructure, access to electricity and/or alkaline minerals will dictate the choice of locations. The correlation between neighboring injection locations has ramifications for the planning, monitoring and governance of different OAE projects, as they will affect each other in downstream coastal areas. For monitoring and verification purposes it will be impossible to disambiguate CO$_2$ drawdown caused by different OAE projects by measurement

alone. Any plans to add alkalinity to the ocean will need to be simulated specifically, ideally with regionally optimized models, and take into account already occurring OAE projects nearby.

### 3.2    CO$_2$ uptake timescales

To measure localized CO$_2$ uptake timescales, we conducted a total of 17 pulse injections (as described in the methods section), placed near all major coastlines. We generally chose locations previously determined as areas of high alkalinity tolerance.

Firstly we observed a very large variety of CO$_2$ uptake timescales (Fig. 4) both in the short term (after 1 year) and in the medium term (after 10 years). After 1 year the uptake fraction $\eta_{CO2} = \Delta DIC / \Delta Alk$ varied between 0.2 and 0.85 and after 10 years most locations resulted in an uptake fraction of 0.65-0.80 consistent with previous work (Tyka et al., 2022; Burt et al., 2021).

A typical behavior is observed for example when releasing alkalinity on the northern coast of Brazil (Fig. 5a). Here the

alkalinity remains long enough at the surface to realize its full CO$_2$ uptake potential within 2-3 years. A number of other tested locations follow this general pattern and are efficient for OAE deployment (Fig. 4a). Here the equilibration follows roughly a single exponential relaxation.

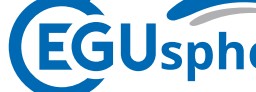

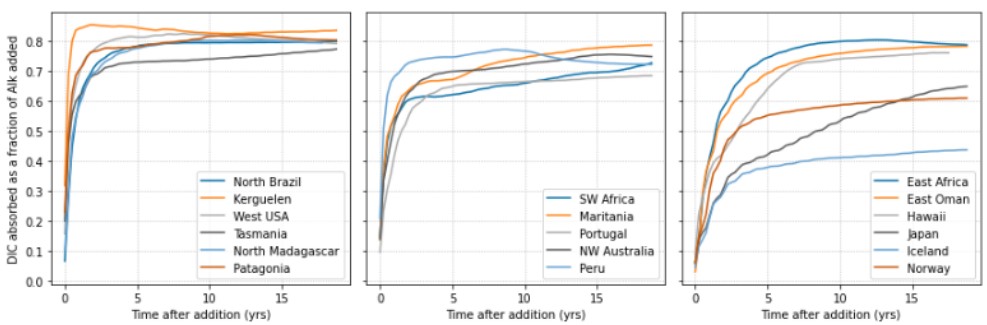

**Figure 4.** $CO_2$ uptake relative to alkalinity addition ($\eta_{CO_2}$) following pulse additions at 17 different locations. (a) Locations which equilibrate fast and reach close to the theoretical maximum of $CO_2$ uptake ( 0.8) (b) Locations which equilibrate fast but reach a lower plateau of relative $CO_2$ absorption (0.6-0.8) with slow further progression. (c) Locations with slow equilibration or significant loss of alkalinity to the deep. Note that in some cases despite the slower initial equilibration high eventual uptake ratios can be achieved eventually.





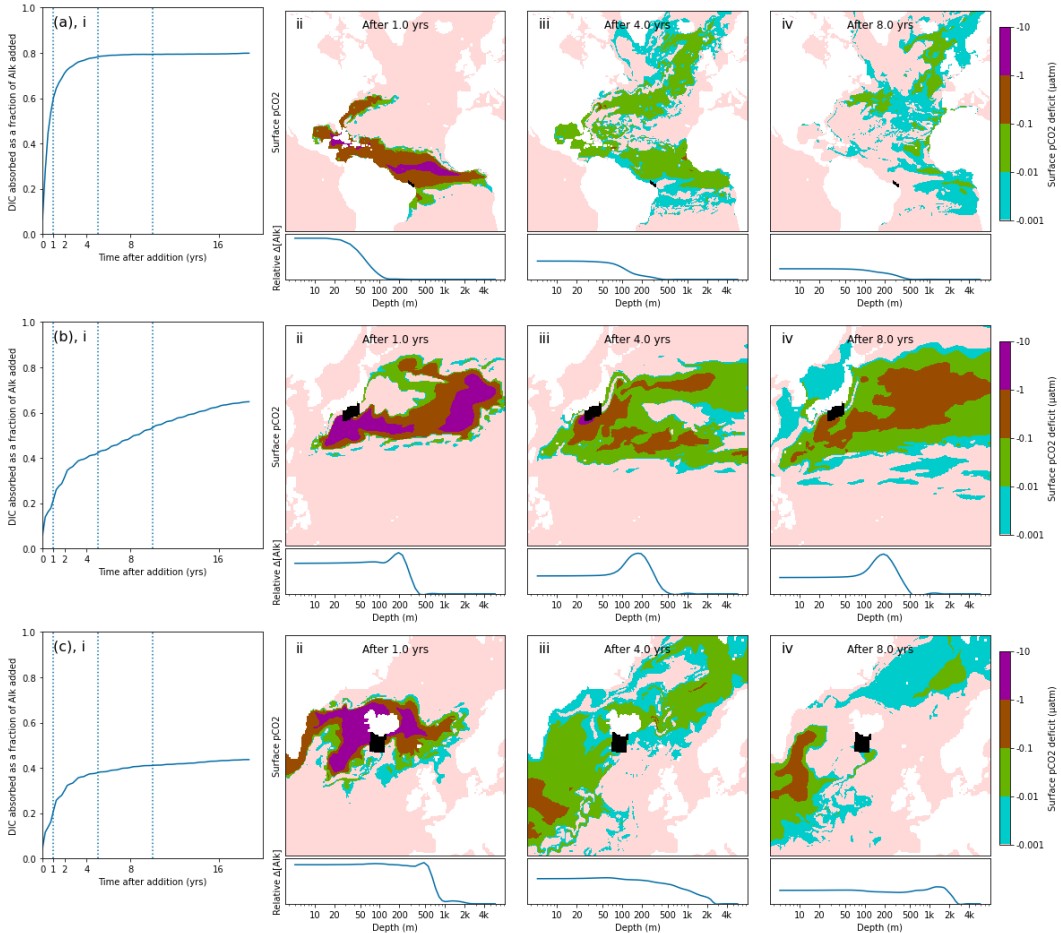

**Figure 5.** Pulse additions of alkalinity in 3 representative locations: (a) Brazil, (b) Japan, (c) Iceland. (i) Excess quantity of $CO_2$ absorbed (relative to the reference simulation) over time following a 1 month pulse injection at various near coast locations, expressed as a molar fraction of the amount of alkalinity added($\eta_{CO_2}$). A significant variation of uptake timescales is observed, depending on speed at which excess alkalinity is removed from surface layers, both in the short term and in the long term. (ii)-(iv) Spatial detail of surface $pCO_2$ deficit and depth residence of excess alkalinity for the same 3 locations shown in A. The initial injection location is indicated by a black area. The surface $pCO_2$ deficit is plotted over time (note the log scale of the colormap), indicating areas which are absorbing extra $CO_2$ (or emitting less $CO_2$) compared to the reference simulation. Below, the relative excess alkalinity is plotted against the depth of the water column (averaged over all lat/lng grid points for each depth). The speed at which alkalinity is lost to the deep in different locations helps explain the difference in the $CO_2$ uptake efficiency achieved. Further locations are shown in detail in Figure S5 in the supplementary material.





Another set of locations appear to lose a significant amount of alkalinity to deeper layers before atmospheric equilibration is achieved. For example injection off the coast of Japan resulted in slow initial uptake as a portion of the alkalinity is subducted

quickly. However in the following decade re-mixing with surface waters gradually returns this alkalinity back towards the surface resulting in slow but steady $CO_2$ uptake (Fig. 5b). Other examples of this delayed $CO_2$ equilibration are shown in (Fig. 4b). These locations exhibit a short mixed layer residence time and have a poor equilibration efficiency (Jones et al., 2014) but equilibration is eventually achieved in the following decades.

Finally some extreme examples of poor long-term equilibration efficiency are found in areas of deep water formation, such

as the northern Atlantic. Here a $CO_2$ uptake ratio of just over 0.4 is achieved and even after 20 years very little further progress is made (Fig. 5c). Around half of the added alkalinity is subducted very deep and will likely remain out of contact with the atmosphere until the global overturning circulation returns these waters to the surface, on the timescale of many hundreds of years. We did not examine the dependence of the time of year (all of our pulses occurred in January) nor were we able to conduct an exhaustive set of locations as was done previously with a coarser global circulation model (Tyka et al., 2022). We

note that for all cases the alkalinity-induced $CO_2$ deficit spreads over a very large area within one year and the changes in $pCO_2$ are in the sub $\mu$atm range. This makes direct monitoring and verification of OAE extremly challanging and will likely need to rely on modelling and indirect experimental verification.

### 3.3 Alkalinity injection from ships

Other than potential ecological impacts on marine life intersecting the caustic release wake of an OAE ship, one concern is

that a short $\Omega_{Arag}$ spike could induce precipitation of $CaCO_3$ (Renforth, 2012). Once nucleated, the $CaCO_3$ particles could continue to grow, even when the pH has returned to normal ocean levels ($\approx$8.1) because the ocean is supersaturated with respect to calcite ($\Omega_{Calc} \approx 2.5 - 6$ ) and aragonite ($\Omega_{Arag} \approx 1.5 - 4$)(Lauvset et al., 2016; Olsen et al., 2017). While the nucleation of $CaCO_3$ is strongly inhibited by the presence of magnesium in seawater (Sun et al., 2015), the growth of existing crystals may not be (Moras et al., 2021). Only once the $CaCO_3$ particle has reached a size and density that causes it to sink would it

stop removing alkalinity from the surface ocean. Thus, depending on the number of particles nucleated, the alkalinity removed from the surface ocean be be larger than the alkalinity added (Moras et al., 2021; Fuhr et al., 2022).

Since the immediate dilution dynamics of alkalinity injected into the wake of a ship is far below the resolution of the ECCO LLC270 global circulation model we examine this process analytically, as described in the methods section. The relevant timescales are compared in Figure 6. In blue are shown the predicted pH and $\Omega_{Arag}$ as a function of time, based on the dilution

formulas given by IMCO (1975) and Chou (1996).

Shown also are homogeneous nucleation times of $CaCO_3$ for comparison. Several studies (Pokrovsky, 1994, 1998) have measured the homogeneous nucleation of $CaCO_3$ in seawater at different saturation states down to $\Omega_{Arag}$=9. Three of these models are plotted in shades of orange and black in Figure 6. Theoretical studies (Sun et al., 2015) suggest that for Mg:Ca ratios of 5.2, as found in seawater, no nucleation of Aragonite occurs at all below $\Omega_{Calc} = 18$ (equivalent to $\Omega_{Arag} \approx 12$), due

to inhibition by Magnesium. This is consistent with Morse and He (1993), however timescales only up to a few hours were examined. Moras et al. (2021) suggested a safe limit of $\Omega_{Arag}$=5 based on alkalinity addition using CaO.





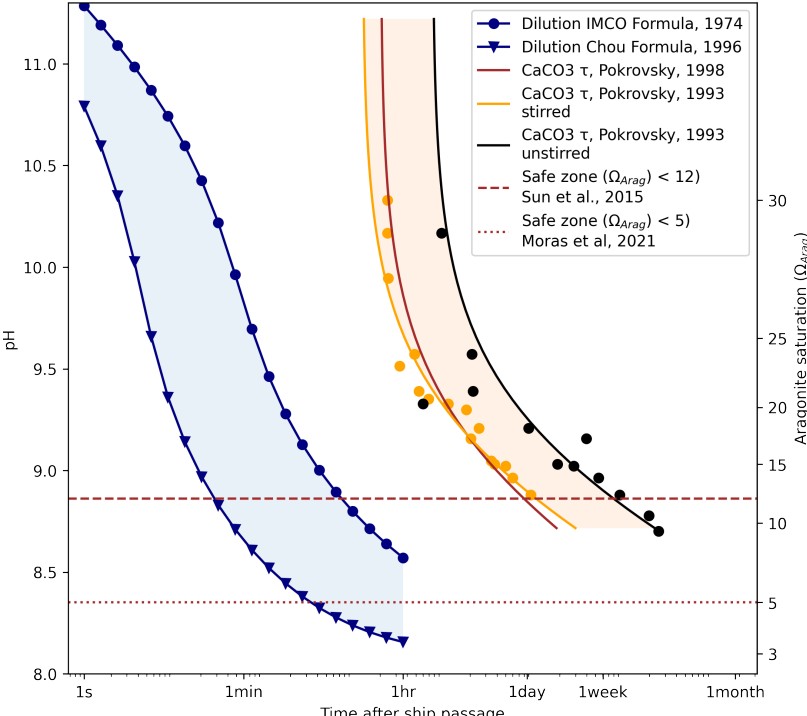

**Figure 6.** The expected time evolution of pH (left scale) and $\Omega_{Arag}$ (right scale) due to dilution for alkalinity injection into a ship wake. Here we assume a large tanker (275m long, 50m wide, traveling at 6m/s) releasing 1.0M NaOH at a rate of 5m$^3$/s. Ships of this size have a capacity of 100 to 200 kilotons of cargo which would take 6-12hrs to discharge. Two previously published dilution models are shown in blue with a large variance apparent. For comparison, the timescales of homogeneous nucleation are also shown (yellow, brown and black). The dashed and dotted lines indicate the estimated $\Omega$ limits for precipitation. Despite the substantial uncertainty in the existing models, dilution can proceed at least 1-2 orders of magnitude faster than precipitation is expected to occur.

Overall it is apparent that ship-wake dilution proceeds at least one order of magnitude faster than the homogeneous nucleation time, thus we can expect that $CaCO_3$ particles will not be induced to nucleate.

At the immediate injection site where the pH exceeds 9.5-10.0 the temporary precipitation of Mg(OH)2 is expected, which
redissolves readily upon dilution (Pokrovsky and Savenko, 1995) and buffers the pH against further increase (not accounted for in Figure 6). The temporary reduction in the Mg:Ca ratio could make $CaCO_3$ nucleation more favorable, but the time spent in this state (<1minute) is likely still well below the required nucleation time.

## 3.4 Transport costs

Alkalinity prepared on land must be transported out to sea, which incurs costs. Having obtained maps of the sustainable rate
of OAE in any given area we can estimate the transport costs for each of our simulated scenarios as described in the methods section. Furthermore, the quantity of alkalinity per tonne depends on the molality of the material moved. For rock-based





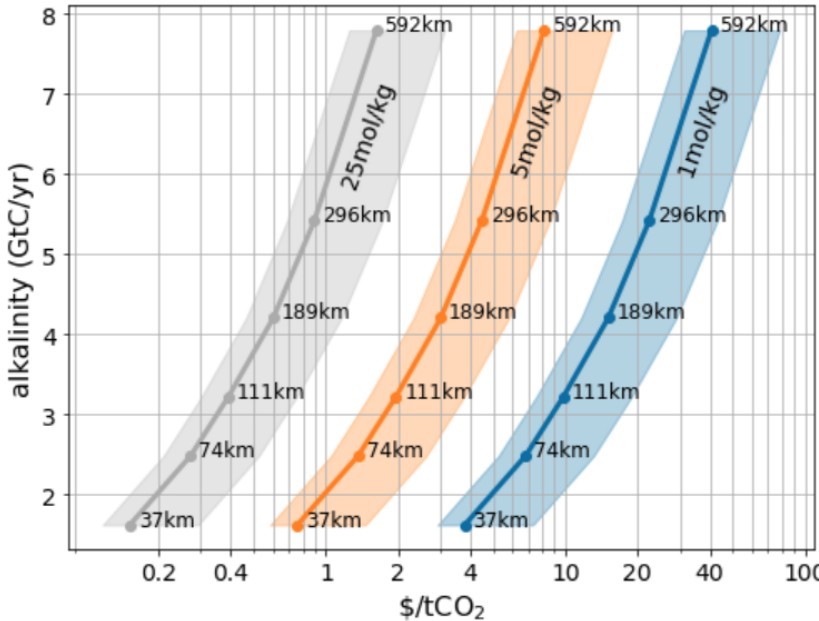

**Figure 7.** Total carbon uptake potential for a variable alkalinity addition strategy that results in a maximal pH change of 0.1 (see Figure 1 for an example) . Carbon uptake is estimated at $\eta_{CO_2} = 0.8$. The strip widths necessary are indicated for each point. Shading denotes the shipping cost uncertainty, which ranges from \$0.0016-\$0.004t$^{-1}$km$^{-1}$ (Renforth, 2012)

methods (Hangx and Spiers, 2009; Schuiling and de Boer, 2011; Renforth, 2012; Montserrat et al., 2017; Rigopoulos et al., 2018; Meysman and Montserrat, 2017), the molality of solid alkaline materials such as olivine is 25mol/kg. For electrochemical alkalinity methods (House et al., 2007; Rau, 2009; Davies et al., 2018; Eisaman et al., 2018; de Lannoy et al., 2018; Digdaya

et al., 2020) that produce alkaline liquids it will be closer to 1kg/mol depending on the industrial processes used and effort spent concentrating the alkaline solution. Figure 7 shows how the transportation costs are influenced by the total desired negative emissions (larger scale requires transport further offshore). For concentrated alkalinity (such as ground rocks) transport costs are not a major contributor to cost, consistent with prior work (Renforth, 2012). However for dilute alkalinity, such as obtained from electrochemical processes, transportation could become a significant contributor to the overall cost. Economic tradeoffs

between the cost of concentrating the alkalinity and the cost of transportation will need to be made.

## 4   Conclusions

In this paper we examined the suitability and effectiveness of near-coast regions of the ocean for alkalinity enhancement (OAE). We conducted a series of medium resolution (0.3°) global circulation simulations in which alkalinity was added to coastal strips of varying width under the constraint of limited $\Delta$pH or $\Delta\Omega_{Arag}$. We found that the resultant steady-state rate

at which alkalinity can be added at any given location exhibits complex patterns and non-local dependencies which vary from



region to region. The allowable injection rate is highly dependent on the surrounding injection pattern and varies over time, responding to external seasonal factors which are not always predictable. This makes it difficult to prevent occasional short spiking beyond the specified limit, thus potentially requiring that the limit is set conservatively in practice. These difficulties are also expected to arise in practice and have repercussions on how such OAE would be performed in reality and how it would be

monitored, regulated and verified (MRV). The non-local nature of the pH effect also likely requires different adjacent countries to coordinate their OAE efforts.

We found that even within the relatively conservative constraints set, most regional stretches of coastline are able to accommodate on the order of a tens to hundreds of megatonnes of negative emissions, with areas with access to fast currents being able to accommodate more, such as East Africa or the coast of Peru. Globally we conclude that near-coastal OAE has the po-

tential to scale to a few gigatonnes of $CO_2$ drawdown, if the effort is spread over the majority of available coastlines. However, given that many other factors will determine suitable locations (such as availability of appropriate alkaline minerals, low-cost energy and geopolitical suitability) the global potential may be lower in practice. We also examine the cost of transport of alkalinity, which increases with global deployment size as the alkaline material needs to be spread over greater distances from the shore. For alkalinity schemes based on dry minerals the transport costs remain minor, but for electrochemical methods,

which produce more dilute alkalinity, this may present limits to scaling.

We also examined the effectiveness and timescale of alkalinity enhancement on uptake of $CO_2$, through pulsed injections and subsequent tracking of surface water equilibration. Depending on the location, we find a complex set of equilibration kinetics. Most locations reach a plateau of 0.6-0.8mol $CO_2$ per mol of alkalinity after 3-4 years, after which there is little further $CO_2$ uptake. The plateau efficiency depends on the amount of alkalinity lost to the deep ocean which will not equilibrate with the

atmosphere until it returns to the surface, on the timescale of 100-500 years or more. The most ideal locations, reaching close to the theoretical maximum of ≈0.8, include north Madagascar, Brazil, Peru and locations close to the southern ocean such as Tasmania, Kerguelen and Patagonia, where the gas exchange appears to occur faster than the surface residence time. The variation of the achievable CO2 drawdown per unit alkalinity on timescales relevant to the climate change crisis and the speed at which equilibration is reached poses further difficulty for verification of CDR credits.

Further study to determine these uptake efficiencies, at a finer location sample resolution and ideally with model ensembles, are needed for optimal placement of OAE deployments. While our results give an overall picture and are indicative of the complexity, more sophisticated biogeochemistry models (e.g, Carroll et al., 2022) and higher-resolution regional coupled biogeophysical models (e.g., Sein et al., 2015; Wang et al., 2022) will be essential for simulation and deployment of real-world OAE projects.

*Code and data availability.*    Code and data will be made available on a publicly available repository upon final publication

*Author contributions.*    JH and MDT designed the experiments, wrote the code, ran the simulations and wrote the paper.



*Competing interests.* The authors declare that they have no conflict of interest.

*Acknowledgements.* We would like to thank Chris Van Arsdale, Lennart Bach, Brendan Carter, Matt Eisaman and Matthew Long for many helpful comments on the manuscript.



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
