# Peer review of "Limits and CO2 equilibration of near-coast alkalinity enhancement"

_EGUsphere, 2022_

## Author Response (AR1)

Point-by-point response to reviewer comments:

**RC1**

Thank you for your in-depth review of our paper ! We've implemented all of the suggestions as detailed below:

- Line 7: " the steady-state OAE rate" is this referring to the rate at which OAE equilibrates with atmospheric $CO_2$? If so possibly consider "the rate at which OAE reaches a steady state…" as this was a difficult sentence to decipher.

    Changed "steady-state OAE" to "sustainable OAE". This is the rate at which alkalinity can be added at a sustained rate while not exceeding the chosen deviation from background pH/Omega/etc.

- Line 9: "…currents allow the…" consider: "currents allow for the…"

    Changed to "currents allow for the…"

- Line 10: "We found that within…" consider " We found that with…"

    Changed to " We found that with…"

- Line 20-21: I found this to be a rather harsh concluding sentence, particularly for an abstract. Furthermore, there was no explanation of how much alkalinity is lost, particularly when in theory we should be able to avoid this loss through our selection of locations (I appreciate that you discuss this sufficiently in your manuscript, however for those readers who only read the abstract this may be misleading). I would recommend briefly expanding on this loss of alkalinity to the deep (e.g. potential for this to occur, the ability for us to avoid this etc.) and moving it up in the abstract so that it is not the concluding sentence.

    Yes, we don't mean to give a gloomy conclusion. These bad locations are in the minority. I've changed the sentence to "Regions of significant downwelling (e.g. around Iceland) should be avoided by OAE deployments, as in such locations up to half of the OAE potential can be lost to bottom waters." I also moved the sentence up, so we end on a more positive note.

- Line 29: "On long geological timescales" this reads oddly and is stating the obvious, I recommend deleting the adjective "long".

    Deleted "long"

- Line 50: Although a relevant study please consider referencing (Guo et al., 2022) https://doi.org/10.5194/bg-19-3683-2022 instead as this study directly looked at the

effects of nickel on phytoplankton.

*Referenced Guo et al.*

- Line 51-52: This seems like an appropriate point to mention the energy costs associated with grinding and therefore benefits to coastal applications using coarser minerals.

  *Rephrased accordingly.*

- Line 70: I think it is important to also state the subsequent drop in $CO_2$ associated with increasing pH as it is not currently clear which variable is impacting organisms (e.g. phytoplankton). Furthermore, although a relevant article, Bach et al. (2019) did not conduct any first-hand research into the ecological effects of OAE. I would recommend including the citation (Subhas et al., 2022) or another ecological study on the effects of OAE to bolster this statement.

  *Added Subhas et al. and added "and a decrease in pCO2, all of which could potentially affect the local ecology" to emphasize the uncertainty in which variable is most relevant.*

- Line 72: The manuscript by Moras et al., 2021 is now published and as such should be referenced as (Moras et al., 2022) https://doi.org/10.5194/bg-19-3537-2022 . I would also recommend adding the citation Hartmann et al., 2022 as done in line 46. This could be said for other sections where Moras et al., 2022 is cited and the inclusion of Hartman et al., 2022 is appropriate.

  *Done throughout the text as suggested.*

- Line 75: delete capital S in "Some"

  *I think instead there's a period missing before "Some". Fixed.*

- Line 83: brackets around "2015" and full stop after et al. not "et al,"

  *Fixed.*

- Line 84: "imagining the distribution of…" imagining is an odd word to have here, consider "simulating".

  *Changed to "simulating"*

- Line 102: again, citing (Guo et al., 2022) https://doi.org/10.5194/bg-19-3683-2022 would be beneficial.

  *Cited Guo et al.*

- Line 104: I think it is important to distinguish the fact that an increase in alkalinity does not necessarily increase DIC. Increases in alkalinity increase the ability of the ocean to sequester $CO_2$ however whether DIC increases or not is dependent on the in-gassing of $CO_2$ from the atmosphere (or alternative method of $CO_2$ injection). I appreciate this is commonly inferred but I believe it is important to highlight this fact so that readers are not under any false impressions about OAE. I recommend deleting "(and subsequent increases in DIC)" or editing this sentence to reflect the above statement.

  Emphasized the difference between OAE and CDR here and throughout the text.

- Line 117-118: "Finally, the effectiveness and timescale of $CO_2$ uptake due to an OAE deployment in a given region is of interest we can define the uptake efficiency $\eta CO_2$ as" this sentence is difficult to read. Consider something along the lines of: "Finally, to assess the effectiveness and timescale of $CO_2$ uptake due to an OAE deployment in a given region of interest we can define the uptake efficiency $\eta CO_2$

  Changed as suggested.

- Line 213: I have one concern/question over the pulse injections of alkalinity. Did these consider the potential for alkalinity to precipitate out at depth? I understand the need to model the potential for high alkalinity/low $CO_2$ water parcels to return to the surface ocean. However, I am concerned that the modelling of such potentially long timescale processes may lead to over/underestimating the return of high alkalinity/low $CO_2$ water parcels to the surface ocean, as it is possible for alkalinity to be removed at depth through precipitation.

  This is a very valid concern and is the subject of study in ongoing work. The carbonate/biology model used in this study is not sophisticated enough to account for such secondary effects on precipitation or secondary effects on biology (such as calcification rate). Given that most precipitation occurs at the surface where oversaturation is greatest and is largely biologically driven by calcifiers, I'm more concerned about the effect at the surface than at depth. On the other hand increased surface calcification could also increase sinking rates and thus transport of organic material to greater depths. These effects are very interesting, likely complex and underscore the need for better models and more experimental work.

- Results heading: The text under the heading "Results" appears to be more of a discussion which also includes the results. Consider changing the heading to "results/discussion" or "discussion" (depending on the journal requirements).

Changed to "Results and discussion"

● Figure 1 caption, line 3: "The variation of sustainable alkalinity flux in different parts of the coastal strip is apparent" I don't think this is necessary for the figure caption, consider moving it to the main body of the text/ incorporating it into the discussion in lines 289-298.

Move to main text.

● Figure 1e and 1f: x-axis labels? Are they simply the number of grid points? I recommend adding at least one x-axis and labels (if both use the same x-axis) to assist the reader.

I assume this is referring to Y-axis labels ? Yes, they are simply counts of grid point count.  I've added the label.

● Figure 1 caption: "...the total global total..." consider changing to "...the total global injection..."

Changed to "The total global injection"

● Figure 1 caption: "the addition rate and pH change stabilize after 5 years" Again I don't think this is necessary. I recommend moving this to the discussion.

Moved to discussion.

● Line 315: consider changing "...can both be found..." to "...can be found, both on the outside and inside of the injection strip"

Changed accordingly.

● Figure 2 caption: again, I would recommend moving the descriptive parts of the figure caption into the main body of the text e.g. sentences "in general regions…" as well as " widening strips allow more…". Much of this is already in the text and is therefore repetitive.

Moved and merged with main text.

● Figure 2B: the axis labels here are odd, consider editing it so that the y-axis labels line up with each other. E.g., 40 is on the same vertical line as 400 etc.

The left and right axes in 2B are different units (as given by the axis labels). The conversion factor between them happens to be close but not exactly 10 and thus they don't line up perfectly: 1mol of Alk absorbs approximated 0.8*12g/mol=9.6 grams of carbon.  The apparent near-conincidence is exacerbated by the log scale.

However, given that we've been using the unit tCO2/yr instead of tC/yr throughout the rest of this paper I've amended this figure to use tCO2/km^2/yr and hopefully that will clear up the confusion.

- Figure 2b caption: change "a large range of injection capacities is observed" to "a large range of injection capacities are observed"

  Sentence deleted according to earlier comment (duplication with main text and excessive figure legends)

- Figure 3; figures seem to be mislabelled. Figure 3c should be 3b I believe. If not please change these around so they are in order (figure 3a, 3b then 3c).

  Yes, thank you for catching that. Labelling was corrected.

- Figure 3 caption: much of the caption is already present in the text, consider deleting the descriptive sections last 2-3 sentences).

  Descriptive sentences merged with main text.

- Figure 3: Consider labelling the figures as; "figure a.i" as done for figure 5.

  Labeled figures as (a,b,c)(i,ii,iii)

- Line 329: should this be "figure 2C", not "figure 3C", also note the text highlighting the locations could be included in the figure 2 caption and then referred to in the text simply as "…the regions depicted in figure 2c…" without the need to relist the countries.

  Figure reference changed and countries list moved to caption.

- Figure 5 caption: "…for the same 3 locations shown in A" should this be "…shown in i"?

  Fixed.

- Line 359-60: Is this in reference to a figure? If so please state the figure.

  Figure reference added.

- Line 381: "be" is repeated, delete

  "Be" deleted.

- Line 392: it isn't clear why this is apparent, if it is based on the figure or from a reference, please clarify this in the sentence.

Clarified by changing to "Figure 7 shows that ship-wake dilution proceeds.."

Supplementary

- figure caption s1, line4: "the total global total injection" change to "the total global injection rate…"

  Changed to "the total global injection rate…"

- figure s4 caption: consider including the general locations of these maps.

  General locations added in caption.

- Figure s5: again, consider adding some general locations. Stating additional specific areas suggests that they have been chosen at random. Consider specifying if these areas were selected at random or why these areas were chosen.

  Added: "The locations were chosen as examples, coarsely distributed along all major coastlines, in order to find and demonstrate the breadth of possible CO2 uptake kinetics." Locations were added as labels on the graphs.

**RC2**

We thank the reviewer for a thorough review and many thoughtful comments which we have incorporated into the manuscript.

1. The choice of pH and omega anomalies as opposed to absolute values when constraining OAE rates: Given the authors justification that the CO2 system constraints they impose on potential OAE are to avoid calcite precipitation, why use pH and omega anomalies as opposed to absolute values? I find this choice problematic given both the log-scale of pH and the non-linear response of the CO2 system (including omega) to alkalinity enhancement. For example, higher pH waters exhibit a smaller change in pH for the same change in [H+] just due to the log-scale.

   While lower omega waters will exhibit a smaller change in omega for the same increase in alkalinity (and the opposite is true of pH). I don't want to necessarily force the authors to rerun computationally expensive simulations but feel this decision to use anomaly thresholds needs to be better justified. Perhaps the authors could assess the extent to which permissible OAE is dependent on background omega and pH values. I also feel that the they are missing a proper comparison between the use of the pH versus omega threshold.

We considered both absolute and relative thresholds as possible experiment designs, but settled on relative thresholds for a variety of reasons, detailed below. We use pH and Omega as two convenient proxies to evaluate the impact of OAE: we want to consider the impact on ecosystems and precipitation.

First let's consider impacts on ecosystems: Here relative changes made more sense to us than absolute ones, since the locally present organisms are likely adapted to the current environment. Imposing an absolute limit everywhere thus doesn't make sense in our opinion. Both pH and Omega affect organisms and thus the relative limit makes sense here.

Next let's consider the use of pH vs [H+] (or [OH-]). In general pH is used as a parameter to describe the ocean state rather than [H+], however the use of [H+] has been suggested before (e.g. Fassbender et al 2021). In our case, the choice of $\Delta$[H+] vs $\Delta$pH depends on whether the actual impact on an organism is proportional to the absolute or relative change in the activity of [H+] (or [OH-]). While this will inevitably depend on each organism, it is reasonable to assume that a relative change is more appropriate. For example, the additional energy expenditure of an organism to maintain its intracellular pH is proportional to the logarithm of the concentration gradient. Thus, to a first approximation, the incurred metabolic cost given a $\Delta$pH is the same for an organism adapted to a pH of 8 as for one adapted to a pH of 7.

As for Omega as a proxy for the avoidance of precipitation, we agree that an absolute limit would capture the true "limit" with respect to precipitation much better than a relative one. However, simulating OAE to such an extent that would raise Omega all the way to its absolute limit everywhere has some concerning drawbacks: The necessary perturbation of [Alk] would be so large in polar regions (where Omega is currently low) that the simulated carbonate system would be very far from where it is currently. It's not clear to us that the ocean state model can be expected to give realistic results since the perturbed state is now far outside the bounds of where the model was parameterized. Nor would we ever expect such a radical addition of Alk to be done in practice, on pH grounds alone (as that limit would be exceeded much sooner).

Therefore we opted to stick with a relative limit and we note that in our simulation the allowed small $\Delta$Omega keeps the absolute Omega below the precipitation limits even in areas (equatorial) where the starting omega is already high. Thus the globally sustainable OAE limits obtained with respect to Omega can be seen as a lower bound on the limit rather than the actual limit itself, which is clearly much higher (from an Omega perspective alone).

We've expanded the methods section to elaborate more deeply on the choices made in our methodology in a new paragraph.

We also noted that we found no correlation between the background pH and the obtained limit (at constrained $\Delta$pH), as the influence of local currents is too dominant a factor. When comparing the alkalinity flux obtained at $\Delta$pH and $\Delta$Omega we find their order of magnitude is highly correlated (consistent with the fact that the limits are largely influenced by current patterns). The correlation is not exact as the relative sensitivity of pH vs Omega wrt Alk (i.e.

∂pH/∂Alk vs ∂Omega/∂Alk) differs, mostly meridionally. We've added sentences to the results section pointing out these properties of the carbonate system.

2. The dependence of results on current climatic conditions: The authors say little about how dependent their results are on current climatic conditions. I would be particularly interested to see how the permissible OAE rates and associated CDR differ under a much higher atmospheric CO2 concentration. Presumably permissible OAE rates are higher (at least when using the delta omega threshold) and the impact on pCO2 (and in turn CDR) will be far greater per unit alkalinity added. Such a simulation may be beyond their scope but I still think some discussion of how their results might change as the anthropogenic carbon content of the ocean increase is required.

We agree this is a fascinating and important, albeit difficult question. We did not explicitly address this for two reasons. Firstly the ocean state model we use (ECCO) was specifically parameterized from contemporary data. Future oceans will potentially change significantly in flow patterns and stratification and thus the application of a contemporary flow field seems inappropriate. Secondly, the anthropogenic emission trajectories are highly uncertain - which future do we even simulate ? Of course multiple IPCC scenarios could be played out however this would increase the scope of this paper very significantly. We feel that future work should address these issues but we believe this is well beyond the scope of the current work. We've added some prose to the discussion section that reflects on these questions and future directions.

3. Figures: The current content and layout of many figures in the manuscript makes for quite painful reading. So much detail is often squeezed into the multi-panel figures that often nothing is legible/understandable without using a high zoom in a pdf

viewer. I suggest the authors simplify their figures, reducing the number of panels and repetitive information. All figure text should conform to a minimum font size.

We simplified each of the figures as suggested in the detailed comments below, increasing font size and reducing duplication by removing panels that are already shown.

Note also that the final double-column figures will utilize the full width of the page, rather than here in this preprint the LaTeX template limited us to 75% of the page width. Thus the fonts will be 35% larger in the final print.

Minor comments

L7-14 The first time I read this I thought you were confusing OAE and CDR. I think you need to introduce the concept of OAE thresholds (and why they may be important) before you mention the steady state OAE rate which most readers (myself included) will assume you are free to choose.

As suggested, we've re-phrased that paragraph, starting the line of thought first with the chosen limitations and only then proceeding to the idea of steady state: "Choosing relatively conservative constraints on $\Delta$pH or $\Delta$Omega, we examine the limits of OAE using the LLC270 (0.3deg) ECCO global circulation model. We find that the sustainable OAE rate vari…"

L21 Isn't it the CDR potential that is lost not the OAE potential? Ie the transport of alkalinity to depth actually facilitates higher surface addition of alkalinity addition without exceeding a given threshold.

Changed to "...as in such locations up to half of the CDR potential of OAE can be lost to bottom waters."

L32 $CO_2$ doesn't dissolve as bicarbonate (although most ocean DIC is in the form of bicarbonate).

Changed to " … and the excess atmospheric $CO_2$ dissolves into the ocean, largely reacting to form (bi)carbonate ions".

L66 "moieties" is not the correct term to use here as it typically refers to part of a molecule. You just mean the sum of the aqueous forms of inorganic carbon.

Changed "carbonate moieties" to "carbonate species"

L91 lat-lon grid?

The grid of points used in the cited study was a latitude/longitude spherical coordinate system. Happy to rephrase this if there is a better way to express succinctly?

L104 avoid "moieties"

Changed "of all chemical moieties involved in the carbonate system" to "all forms of CO2 in the carbonate system"

L119-120 provide the units of DIC and Alk used otherwise it's unclear that this is a molar ratio.

Added "ηCO2 is a unitless molar ratio."

L121-122. Can the authors say more here about this 0.8 value and what effects it?

"The exact value depends on the parameters of the carbonate system, i.e. Alk, DIC, temperature etc., with a typical range of 0.75-0.85". The local surface variation was explored in Tyka et al 2022."

L144-145 The movement of DIC is controlled by the ocean physics not these 5 biogeochemical tracers which are presumably used to compute the ocean CO2 system.

Changed "...uses 5 biogeochemical tracers [..] to simulate the movement of total dissolved inorganic carbon (DIC) within the ocean" to " …uses 5 biogeochemical tracers [..]  to simulate the carbonate system."

L147 Is this freshwater flux from rivers?

Yes, it is from rivers and from rainfall.

L150 uatm are units of partial pressure. Do you hold the concentration or partial pressure of atmospheric CO2 constant ie can local changes in atmospheric pressure influence gas exchange?

We hold the partial pressure constant, not the absolute concentration.

L154 More detail on how the imported wind speeds are used to calculate gas exchange would be useful here.

We included a specific reference to the way gas-transfer is calculated: "Wind speeds, used to calculate the gas exchange, are imported from the LLC270 forcing data and the air-sea exchange of CO2 is parameterized with a uniform gas transfer coefficient [Wanninkhof, 2018]."

L234 Any rationale for January pulses?

No particular rationale. We only had enough computing time to run a handful of these pulse simulations so we had to pick a time point.

L240-241 Not sure this sentence should be a distinct paragraph.

Merged with the following paragraph.

L265-277 Perhaps make it clear that you're only considering the financial and not carbon cast of transport.

We changed the first sentence to "Alkalinity prepared on land must be transported out to sea, which adds to the total cost of the achieved negative emissions (in \$/tCO2)." Of course reducing the shipping load also reduced CO2 emissions associated with that shipping.

L278 Results "and discussion" – there is a lot more than results in this section.

Changed to "Results and Discussion"

Figure 1. A very busy figure. Fonts are too small to be legible. Axes labels are missing on panels e/f/g.

Removed panels e/f/g as they are replicated in supplementary figures which increased the size of figure, especially fonts. Y Axes labels (histogram counts) were added in the supplementary figures.

Figure 2. Another very difficult figure to read. Suggest reducing the number of panels and enhancing font size. Some alkalinity flux axes labels for are missing in panels a/b.

We split the future into two figures and fixed the axes labels.

Figure 2 legend, Line 2- I would call this CDR potential not OAE potential Given the different research questions this manuscript addresses I think this distinction needs to be clear.

This sentence was removed from the figure legend and worked into the text as suggested by Reviewer 1. However, we agree that the distinction is critical and we have tried to make the distinction between OAE and CDR clearer throughout the text.

Figure 3. Panel labels b/c are incorrectly ordered. Panels e/f/g are not interpretable at their current size.

Panel label order fixed. Increased the size of panels e/f/g and increased font sizes overall.

L326 Largest per unit area fluxes. The globally aggregated fluxes are shown to be higher for wider strips.

My understanding is that the meaning of "flux" is already area normalized, i.e. a flux is "substance flow rate per unit area".

L331-332 This sentence is unclear, please clarify.

Hopefully clarified by changing and expanding the sentence to "In general we find that the sensitivity of pH and ΩArag with respect to the local flux of alkalinity highly dependent on the surrounding pattern of injection. In other words the limit at which alkalinity can be added at a given location depends on the alkalinity addition rate at neighboring locations, up to some distance.

L335 Unclear where this 25% comes from. Fig 3e/f/g appear to show 8% and 30% declines in the 200km and 400 km simulations.

"Placing injection patches 400 km apart instead of 200 km apart did not further increase the sustainable flux in each injection patch, and reduced the overall injection capacity by 25%". This sentence is comparing panels f and g, not e and f. In other words g is 25% lower than f. (233 Tmol/yr / 312 Tmol/yr = 0.75)

L341. Do the authors expect this length scale to vary considerably regionally?

My sense is that yes, this length scale is non-isotropic and will vary from place to place. Indeed Jones et al. 2016 reports "The global median spatial autocorrelation (e-folding) length is 400 ± 250 km, with large variability across different regions."

We amended the sentence at L341 to indicate the variability ".. injection areas should be placed at most 200-400 km apart, however the optima will depend on the local current patterns."

L351-352 clarify these are molar ratios.

Changed sentence beginning to "After 1 year the molar uptake fraction.."

Figure 4. Reiterate the units are mol/mol.

Added "(molar ratio)" to figure legend.

Figure 5. Another busy figure. The "surface pCO2" label is unnecessary and conflicts with the surface pCO2 deficit label.

We've removed the extraneous "surface pCO2" label and will increase the font size to match that of the main text in the final layout of the paper.
We agree the figure is busy, but we believe this is justified as we're trying to visually show

the interplay between spread of the deficit, the CO2 uptake behavior and the depth-loss of alkalinity. We chose three scenarios to exemplify the breath of behaviors possible.
This figure is intended to utilize the full width of the page in the final print, rather than 75% as in this preprint.

Figure 5. legend. I think some of this legend may be misleading. Presumably change in the surface pCO2 over time is also influence by mixing/circulation/biology (and not only gas exchange).

Reviewer 1 suggested moving interpretations and discussions from figure legends to the main text (to avoid duplication). Thus we have moved discussion of effects other than gas exchange into the main text.

L371 Some local deficits >1uatm are shown in Figs 5bii and 5cii.

We amended the paragraph to better reflect the intended point:

"We note that for all cases the alkalinity-induced CO2 deficit spreads over a very large area within one year and a significant fraction of the CO2 uptake occurs after the deficits have diluted to the sub µatm range. This makes direct monitoring and verification of OAE extremely challenging and will likely need to rely on modeling and indirect experimental verification."

L405 mol/kg?

Changed to mol/kg.

L413 I would call this a high-resolution global ocean model (particularly compared to most global obgc models and papers that have simulated OAE previously).

Changed "medium" to "high"

L421 Shouldn't the R in MRV be for Reporting?

Changed to "reported".

---

## Author Response (AR2)

Dear Dr. Tyka:

Thank you for submitting your revised version. I have read it with pleasure, and I am happy to inform you that your paper is now accepted pending some minor technical corrections.

With best regards, Jack Middelburg, Associate editor
—————————————————————————————————————————————————————————————

Dear Dr. Middelburg,

Thank you for your comments on our manuscript - I'm delighted that you enjoyed it! I've corrected all the issues highlighted, plus a few punctuation errors we found.
I also added different line styles to figure 5 to make the different lines more distinguishable for colorblind readers.

With best regards,

 Mike

—————————————————————————————————————————————————————————————
All through: please check the proper use of subscript for CO2 (including the title), lines 71, 387, 388, 467

All occurrences of $CO_2$, $pCO_2$ or $\eta CO_2$ have been properly subscripted.

Line 49: it is nickel in text of Ni as abbreviation.
Changed to lowercase "nickel"

Line 91: please use another term than finer-grained because some of the reader may relate this to grain-size of olivine or other solid phase alkaline substances.
Changed "fine-grained"  to "more spatially resolved"

Line 147: bracket missing around 2005
Fixed.

Line 166: brackets missing around 2017
Fixed.

Line 243: closing bracket is missing
Fixed.

Line 407: …surface ocean can be larger….
Corrected.

Line 416: aragonite
Changed to lowercase.

Line 417: magnesium
Changed to lowercase.

Line 421: subscript for Mg(OH)2
Subscript fixed.